# On the inability of Gaussian process regression to optimally learn compositional functions

**Matteo Giordano**[*]
Department of Statistics
University of Oxford
matteo.giordano@stats.ox.ac.uk

**Kolyan Ray**[*]
Department of Mathematics
Imperial College London
kolyan.ray@imperial.ac.uk

**Johannes Schmidt-Hieber**[*]
Department of Applied Mathematics
University of Twente
a.j.schmidt-hieber@utwente.nl

## Abstract

We rigorously prove that deep Gaussian process priors can outperform Gaussian process priors if the target function has a compositional structure. To this end, we study information-theoretic lower bounds for posterior contraction rates for Gaussian process regression in a continuous regression model. We show that if the true function is a generalized additive function, then the posterior based on any mean-zero Gaussian process can only recover the truth at a rate that is strictly slower than the minimax rate by a factor that is polynomially suboptimal in the sample size $n$.

## 1  Introduction

Hierarchical methods are at the heart of modern machine learning and provide state of the art performance for a variety of different problems. While the role of the depth in deep neural networks has been extensively investigated [37, 44, 45, 24, 33], the theory underlying deep Gaussian process priors is still in its infancy. In this work we identify a structured recovery problem for which deep Gaussian process priors are known to achieve near optimal behavior. By proving that for this problem any Gaussian process prior will lead to a sub-optimal posterior shrinkage behavior, we provide a theoretical justification for the use of deep methods for Bayesians.

Gaussian process (GP) priors are arguably the most widely used Bayesian nonparametric priors in machine learning, having found success in multiple settings [26]. Reasons for their popularity include their interpretability, ability to flexibly model target functions, model spatial or temporal correlations, incorporate prior knowledge such as stationarity, periodicity or smoothness, and their provision of uncertainty quantification. This work establishes information-theoretic bounds on the inability of GPs to optimally model generalized additive models, thereby quantifying certain limitations of GP methods.

We will study the performance of GPs in a continuous analogue of nonparametric regression via quantifying the speed of posterior concentration/contraction around the true regression function. This is a frequentist assessment that measures the typical behaviour of the posterior distribution under the true generative model. The performance of Bayesian methods is known to be sensitive to the choice of prior, particularly in high- and infinite-dimensional settings [11, 20], and it is therefore

---

[*]All authors contributed equally.

36th Conference on Neural Information Processing Systems (NeurIPS 2022).

crucial to understand under what conditions and prior calibrations Bayesian inference is reliable. The frequentist analysis of Bayesian methods has become a standard tool for such tasks in both the statistics and machine learning literatures (e.g. [36, 25, 43, 42, 6, 27, 28]), see the monograph [12]. Precise definitions and further discussion are provided below.

If the regression function is a generalized additive model, it has been shown that deep neural networks with properly chosen network architecture achieve the optimal minimax rate of convergence [32]. We show here that for this structural constraint, *any* mean-zero Gaussian process, irrespective of the choice of covariance kernel, will concentrate around the truth at a rate that is strictly slower than the minimax rate by a factor that is polynomially suboptimal in the sample size $n$. Thus any posterior GP places a significant portion of its posterior probability on functions that are further from the truth in $L^2$-distance than the minimax rate of estimation, which has negative implications for both Bayesian estimation and uncertainty quantification.

In contrast, it has recently been shown that deep Gaussian processes [23, 8], which consist of iterations of Gaussian processes and can be viewed as a Bayesian analogue of deep neural networks, are able to attain the minimax rate of contraction for arbitrary compositional classes [10]. As these compositional classes contain generalized additive models as a special case, we have shown that deep GPs provably outperform GPs from a statistical perspective and should thus be preferred.

In related work, it has been established that (not necessarily Bayesian) linear estimators can only attain polynomially suboptimal *linear estimation rates* over certain spatially inhomogeneous Besov classes [9] or for functions with discontinuities [21]. Agapiou and Wang [1] use the results of [9] to show that GPs also cannot concentrate faster than the linear rate for such Besov classes, see also [31] for some related examples. Castillo [4] establishes lower bounds for concentration rates in terms of abstract conditions on the 'concentration function' of the GP, which can be verified for certain specific GP choices. The lower bound ideas of [9, 21] have also been extended to various spatially inhomogeneous and discontinuous functions settings to demonstrate the theoretical advantages of deep learning over linear estimators [15–17, 34, 35, 38]. Finally, for generalized additive models, [32] shows that (possibly nonlinear) wavelet methods can only achieve suboptimal rates.

Most of the above works exploit the results from [9, 21] showing that minimax rates for linear estimators are slower than full minimax rates in these settings. Since linear rates are unavailable for generalized additive functions, and seem difficult to prove using existing techniques, we instead use a different and novel proof approach. We exploit the specific structure of the GP posterior mean in regression to construct a direct lower bound for its prediction risk. In an effort to develop the new tools needed to solve such problems, we further provide a second proof in the supplement that involves first reducing the regression setting to a simplified sparse sequence model in which we derive linear rates. While these do not imply linear rates in the full model we consider, they prove sufficient to construct lower bounds for arbitrary GP posterior means.

## 2 Main results

### 2.1 Problem setup and posterior contraction rates

In the usual multivariate nonparametric regression model we observe $n$ training samples $(X_1, Y_1), \ldots, (X_n, Y_n)$ arising as

$$Y_i = f(X_i) + w_i, \qquad w_i \sim^{iid} N(0, 1), \tag{1}$$

where the design points $X_i$ are either independent and uniformly distributed random variables on $[0, 1]^d$ or equally spaced lattice points in $[0, 1]^d$. To simplify technical arguments due to the discretization and provide a clearer exposition, a standard approach in statistical theory is to consider instead the 'continuous' analogue of this model [19]. The discrete model (1) is asymptotically equivalent [30] (as $n \to \infty$) to observing a realisation $Y = Y^n = (Y_x : x \in [0, 1]^d)$ of the multidimensional Gaussian white noise model

$$dY_x = f(x)dx + \frac{1}{\sqrt{n}}dW_x, \qquad x \in [0, 1]^d, \tag{2}$$

where $W = (W_x : x \in [0, 1]^d)$ is a $d$-dimensional Brownian motion. For large $n$, the two models thus behave identically from a statistical perspective.

**Notation**: For a domain $D$ and $1 \leq p \leq \infty$, denote by $L^p(D)$ the space of all measurable functions $f : D \rightarrow \mathbb{R}$ that have finite norm $\|f\|_{L^p(D)} := (\int |f(u)|^p du)^{1/p}$ if $p < \infty$, and are essentially bounded on $D$ if $p = \infty$. If there is no ambiguity, we write $L^2$ for $L^2[0,1]^d$ and $\langle f, g \rangle_2 = \int_{[0,1]^d} f(u)g(u)du$ for the corresponding inner product. Let $P_f = P_f^n$ denote the probability distribution of $Y = Y^n$ arising from (2) with corresponding expectation $E_f$.

We consider the Bayesian setting where we assign to $f$ a (possibly $n$-dependent) mean-zero Gaussian process prior $\Pi = \Pi_n$ on $L^2[0,1]^d$ with covariance kernel $K$, i.e. $\Pi$ is a (Borel) Gaussian probability measure on $L^2[0,1]^d$ such that for $f \sim \Pi$, we have

$$E f(x) = 0, \qquad E[f(x)f(y)] = K(x,y), \qquad x, y \in [0,1]^d. \qquad (3)$$

The posterior $\Pi_n(\cdot|Y)$ is then computed as usual using Bayes formula and is again a GP by conjugacy, exactly as in the discrete model (1) [26], see (7) for the exact expression. We make the following frequentist assumption:

**Assumption 1.** *There is a true $f_0 \in L^2[0,1]^d$ generating the data $Y \sim P_{f_0}$ according to (2).*

We study the behaviour of the posterior distribution $\Pi_n(\cdot|Y)$ under Assumption 1, in which case it can be treated as a random probability distribution whose (frequentist) randomness depends on $Y$. We next introduce the notion of uniform posterior contraction.

**Definition 1.** *We say that the posterior contracts about $f_0$ at rate $\varepsilon_n \rightarrow 0$ uniformly over a sequence of classes $\mathcal{F}_n \subset L^2[0,1]^d$ if as $n \rightarrow \infty$*

$$\sup_{f_0 \in \mathcal{F}_n} E_{f_0}\Pi_n(f : \|f - f_0\|_{L^2} \geq \varepsilon_n|Y) \rightarrow 0.$$

Posterior contraction is often stated in the equivalent form

$$\Pi_n(f : \|f - f_0\|_{L^2} \geq \varepsilon_n|Y) \rightarrow 0 \qquad \text{in } P_{f_0}\text{-probability}$$

as $n \rightarrow \infty$, which says that the posterior puts all but a vanishingly small amount of probability on $L^2$-balls of shrinking radius $\varepsilon_n$ about the true $f_0$ generating the data [12]. We are typically interested in the size of the *smallest* such $L^2$-ball, i.e. the fastest rate $\varepsilon_n \rightarrow 0$ such that Definition 1 holds. Such results not only quantify the typical distance between a point estimator (e.g. posterior mean or median) and the truth, but also the typical spread of the posterior about the truth. Ideally, most of the posterior probability should concentrate on a ball centered at the true $f_0$ with radius proportional to the *minimax* estimation rate, see Section C.1 for more details. Indeed, since posterior contraction at rate $\varepsilon_n$ automatically yields a point estimator with convergence rate $\varepsilon_n$ ([12], Theorem 8.7), the minimax rate provides an information-theoretic lower bound for $\varepsilon_n$. Posterior contraction at a fast rate is a necessary condition for statistically good Bayesian estimation and uncertainty quantification.

We use the $E_{f_0}$-expectation formulation of contraction to quantify the *uniformity* over $f_0 \in \mathcal{F}_n$ needed to make our lower bounds precise and meaningful. Such uniformity captures that the true $f_0$ is unknown and rules out unrealistic and trivial situations, such as taking a GP with prior mean equal to the true $f_0$ and arbitrarily small covariance. Since we consider *worst case* (uniform) rates in Definition 1 and it is well-known that the covariance kernel $K$ of the GP determines the characteristics of the corresponding GP [26], including its posterior contraction rates [40], we without loss of generality restrict to mean-zero GPs.

There is by now an extensive literature on proving *upper bounds* for contraction rates [12], including for GPs [40]. These contraction rates are usually uniform over suitable function classes $\mathcal{F}_n$, although the results might not be formulated in this way in the literature.

## 2.2 Lower bounds for posterior concentration in Gaussian processes regression

While the classical approach in nonparametric statistics is to make smoothness assumptions on the regression function $f_0$, recent work in the deep learning literature has studied compositional assumptions. A special case are generalized additive models of the form

$$f_0(x_1, \ldots, x_d) = h\Big(\sum_{i=1}^{d} g_i(x_i)\Big)$$

with $g_1, \ldots, g_d, h$ univariate and Lipschitz continuous functions such that all Lipschitz constants are bounded by $\Lambda$ and $\|g_1\|_{L^\infty[0,1]}, \ldots, \|g_d\|_{L^\infty[0,1]}, \|h\|_{L^\infty(\mathbb{R})} \leq \Lambda$. The class of all functions of this form is denoted by $\mathcal{G}(\Lambda)$ and we define $\mathcal{G} = \mathcal{G}(1)$. Generalized additive models are among the most popular structural constraints in function estimation [14].

Equation (13) in [32] and the subsequent discussion shows that the minimax estimation rate $R_n(\mathcal{G}(\Lambda))$ satisfies

$$cn^{-1/3} \leq R_n(\mathcal{G}(\Lambda)) \leq C(\log n)^3 n^{-1/3}$$

for some $C, c > 0$. Up to logarithmic factors, this rate is obtained by carefully calibrated deep GP priors that first assign a hyperprior to different compositional structures and subsequently put GP priors on all functions in the function composition [10].

We are now ready to state the main lower bound results of this paper, which we emphasize applies to *any* GP, irrespective of the choice of covariance kernel (which may itself depend on $n$).

**Theorem 1.** *Let $\Pi_n$ be any sequence of mean-zero Gaussian process priors on $L^2[0,1]^d$. For any $0 < \delta < 1/4$ and $n \geq N(d, \delta)$ large enough, the corresponding posterior distributions satisfy*

$$\sup_{f_0 \in \mathcal{G}} E_{f_0} \Pi_n(f : \|f - f_0\|_{L^2} \geq C_d n^{-\frac{2+d}{4+4d}} |Y) \geq 1/4 - \delta,$$

*where $C_d = \frac{1}{10 \cdot 2^d} \left( \frac{1}{2(d+2)!} \right)^{\frac{d}{4+4d}}$. In particular, if the posterior distributions contract at rate $\varepsilon_n$ uniformly over $\mathcal{G}$, that is*

$$\sup_{f_0 \in \mathcal{G}} E_{f_0} \Pi_n(f : \|f - f_0\|_{L^2} \geq \varepsilon_n |Y) \to 0$$

*as $n \to \infty$, then $\varepsilon_n > C_d n^{-\frac{2+d}{4+4d}}$.*

For dimension $d \geq 3$, the above rate is strictly slower than the minimax rate $n^{-1/3}$ by the polynomial factor $n^{\frac{d-2}{12+12d}}$. As the domain dimension $d$ tends to infinity, our lower bound is approximately $n^{-1/4}$, in which case Theorem 1 says the GP posterior will place at least probability $1/4 - \delta$ on posterior draws $f \sim \Pi_n(\cdot|Y)$ having suboptimal reconstruction error $\|f - f_0\|_{L^2} \gtrsim n^{-1/4}$. In contrast, posteriors based on properly calibrated deep GPs will place almost all their probability on draws with minimax optimal reconstruction error of order $n^{-1/3}$ [10]. Thus GP posteriors provide poor reconstructions of $f_0 \in \mathcal{G}$ compared to deep GPs.

Theorem 1 says that no GP can attain a faster rate than $n^{-\frac{2+d}{4+4d}}$. It does not say that a given GP even attains this suboptimal rate, and specific GPs can perform much worse than this bound. We in fact conjecture that our lower bound can be improved to show that even this rate is not attainable by any GP, see Section 2.3 for more discussion. Nonetheless the main message from Theorem 1 is already clear, namely that GPs underperform in this setting by an already significant factor that is polynomial in the sample size $n$.

The last result says that generic posterior draws will be poor reconstructions of $f_0$ with significant posterior probability. The same holds for the posterior mean.

**Theorem 2.** *Let $\Pi_n$ be any sequence of mean-zero Gaussian process priors on $L^2[0,1]^d$. Then the corresponding posterior means $\bar{f}_n = E^{\Pi_n}[f|Y]$ satisfy for $n \geq 2(d+2)!$,*

$$\sup_{f_0 \in \mathcal{G}} E_{f_0} \|\bar{f}_n - f_0\|_{L^2} \geq C'_d n^{-\frac{2+d}{4+4d}},$$

*where $C'_d = \frac{1}{2^{d+1}} \left( \frac{1}{2(d+2)!} \right)^{\frac{d}{4+4d}}$.*

Theorem 2 states that the posterior mean $\bar{f}_n$ will also be a suboptimal reconstruction. Since the posterior is also Gaussian by conjugacy, and hence centered at $\bar{f}_n$, this captures the notion that posteriors draws from the 'center' of the posterior will also be poor - this is thus generic behaviour and not an artifact of unusual behaviour in the posterior tails.

Our results also have implications for Bayesian uncertainty quantification (UQ) through the use of credible sets, which is one the main motivations for using GPs. Bayesian credible sets are often

used as frequentist confidence sets, a practice which can be rigorously justified in low dimensional settings ([39], Section 10.2), but is much more subtle in high-dimensional settings like GPs [11]. Whether a resulting credible set contains the truth will depend on the size of the posterior standard deviation relative to the distance between the posterior mean and the truth, which Theorem 2 says is greater than the minimax rate with high-probability. In the best case scenario, the former is greater than the latter and thus the credible set will provide honest and reliable UQ. However, in this case the posterior standard deviation must be at least of the order of the contraction rate, and hence the resulting credible sets will be (polynomially) much too large and uninformative. In the worse case, when the posterior standard deviation is smaller, the credible sets will not contain the truth and provide misleading (useless) UQ.

Note that we are considering UQ for the *entire* function $f_0$ - it may still be possible for the GP posterior to provide efficient UQ for certain low-dimensional functionals of interest, which is itself a current topic of research [5, 29].

The proofs are based on a number of inequalities that are easy to state. Relying on an exponential concentration inequality and Anderson's inequality, we show that Theorem 1 follows from the lower bound for the posterior mean in Theorem 2 (cf. Section 4.1). For Theorem 2 we provide two different proofs. In the first, we lower bound the $L^2$-risk of the posterior mean over arbitrary finite subsets $\{f_1, \ldots, f_m\} \subset L^2[0,1]^d$ (cf. Lemma 7), with the lower bound expressed in terms of expansions of the functions $f_i$ in the basis coming from the Karhunen-Loève expansion of the GP. We then construct a suitable collection of orthogonal generalized additive models for which the lower bound is maximized *independently* of the choice of GP basis, as is needed to deal with arbitrary GPs (cf. Section 4.3). Given a *specific* choice of GP, our lower bound techniques can yield more refined results by tailoring the alternatives $f_1, \ldots, f_m$ to be especially poorly aligned (difficult to learn) for that specific covariance function, see Section 2.3.

In a second proof of Theorem 2, presented in Section B in the supplement, we reduce the problem of studying orthogonal generalized additive models to the sub-problem of studying the risk of linear estimators in a one-sparse sequence model (cf. Section B.1). For the latter, we exactly compute the linear minimax risk in Lemma 9, which, by exploiting the orthogonality, in turn implies the lower bound in Theorem 2. This second proof sheds additional light on the limitations of GP regression by furnishing new tools to study problems that can be connected to sparse models.

## 2.3 A sharper lower bound for a specific class of Gaussian processes

While Theorem 1 holds for arbitrary GPs, we now show that our methods can yield sharper lower bounds in the case of specific GPs, implying an even stronger suboptimality in the associated reconstruction error. We illustrate this by considering Gaussian wavelet series priors. Wavelets have been extensively used in Bayesian nonparametrics to construct priors in function spaces, including applications in inverse problems [41, 2], and image and signal processing [3, 18] among the others; see e.g. [22] for an overview.

Let $\{\psi_\gamma, \ \gamma \in \Gamma\}$, with $\Gamma = \{\gamma = (j,k) : j = -1, 0, 1, \ldots, \ k \in I_j\}$, be a compactly supported wavelet basis of $L^2(\mathbb{R})$ restricted (with boundary correction) to $[0,1]$, cf. [7] or also [13, Chapter 4]. The collection $\{\psi_\gamma, \ \gamma = (\gamma_1, \ldots, \gamma_d) \in \Gamma^d\}$, with $\Gamma^d = \Gamma \times \cdots \times \Gamma$ ($d$ times) and $\psi_\gamma(x_1, \ldots, x_d) := \prod_{i=1}^d \psi_{\gamma_i}(x_i)$ provides a tensor product wavelet basis of $L^2[0,1]^d$.

Consider a mean-zero Gaussian wavelet series prior on $L^2[0,1]^d$ with Karhunen-Loève expansion

$$f(x) = \sum_{\gamma \in I} \sqrt{\lambda_\gamma} \xi_\gamma \psi_\gamma(x), \qquad x = (x_1, \ldots, x_d) \in [0,1]^d, \qquad (4)$$

for some arbitrary index set $I \subseteq \Gamma^d$, coefficients $\lambda_\gamma > 0$ and $\xi_\gamma \sim^{iid} N(0,1)$. Compared to the proof of the general lower bound in Theorem 1, fixing a specific GP basis allows to consider compositional functions that are especially difficult for that GP to approximate. For wavelet bases, such approximation theory was developed in [32], which we employ to obtain the next lower bound.

**Theorem 3.** *Let $\Pi_n$ be any sequence of mean-zero Gaussian wavelet series priors on $L^2[0,1]^d$, defined as in* (4) *with possibly $n$-dependent $\lambda_\gamma > 0$. For any $0 < \delta < 1/4$ and $n \geq N(d, \delta)$ large enough, the corresponding posterior distributions satisfy*

$$\sup_{f_0 \in \mathcal{G}} E_{f_0} \Pi_n(f : \|f - f_0\|_{L^2} \geq Cn^{-\frac{1}{2+d}} | Y) \geq 1/4 - \delta,$$

*where $C = C(d, \psi) > 0$ only depends on $d$ and the wavelet basis. Furthermore, the corresponding posterior means $\bar{f}_n = E^{\Pi_n}[f|Y]$ satisfy for all $n$*

$$\sup_{f_0 \in \mathcal{G}} E_{f_0} \|\bar{f}_n - f_0\|_{L^2} \geq C' n^{-\frac{1}{2+d}},$$

*where $C' = C'(d, \psi) > 0$ only depends on $d$ and the wavelet basis.*

Thus, for the class of Gaussian wavelet series priors, Theorem 3 provides a sharper lower bound than Theorem 1, as $n^{-\frac{1}{2+d}} \gg n^{-\frac{2+d}{4+4d}}$ for all $d$. Furthermore, the former rate deteriorates as $d$ grows, showing that the performance of Gaussian wavelet priors suffers from a form of curse of dimensionality. In particular, when $d$ is large, the fastest rate $n^{-\frac{1}{2+d}}$ achievable by Gaussian wavelet priors is much slower than the minimax rate $n^{-1/3}$ achieved by deep GPs.

## 3 Discussion

In this work, we showed that no mean-zero Gaussian process can attain the minimax estimation rate over a class of generalized additive models. We proved this by developing new tools for lower bounding posterior contraction rates for GPs. We provided two different proofs, each of which sheds different light on these phenomena.

It remains open whether even sharper lower bounds can be obtained. In this context, an important question is whether all GPs suffer from a curse of dimensionality in their posterior contraction rate, which we showed was incurred for a specific wavelet-based GP. Our lower bound $n^{-(2+d)/(4+4d)}$ for arbitrary GPs in Theorem 1 is never slower than $n^{-1/4}$, irrespective of the domain dimension $d$. While the obtained lower bound already demonstrates the suboptimality of GPs by a polynomial factor in $n$, say when compared to deep GPs, our stronger lower bound $n^{-1/(2+d)}$ for Gaussian wavelet series priors becomes arbitrarily slow with increasing $d$. We thus proved that for this subclass of GPs, a stronger curse of dimensionality is unavoidable

To avoid the curse of dimensionality, a common two-stage approach is to first map the input features to a learned lower dimensional space and then apply Gaussian process regression to these new features. Our results do not apply to this scenario, in particular when the feature extraction map can be be nonlinear, since the transformed underlying function may no longer be compositional in terms of the new learned features.

We expect similar results to hold for non-uniformly distributed input variables and heteroscedastic noise. Let $\sigma$ be a positive function and $\rho$ a density on $[0, 1]^d$. If $\rho, \sigma$ are bounded away from zero and infinity, we conjecture that, up to constants, the same lower bounds can be derived working in the more general nonparametric regression model (1) with, conditionally on $X_i$, independent noise variables $w_i|X_i \sim N(0, \sigma^2(X_i))$ and $X_i \sim^{iid} \rho$.

Ideally, one wants to develop a broader understanding of which structures in the data-generating distribution or regression function (deep) Gaussian processes can or cannot adapt to. In order to understand the advantages of deep learning there is an extensive theoretical literature comparing shallow to deep nets. More refined versions show that for certain approximation rates a minimal network depth is necessary [24]. To gain some insights into the role of depth in deep Gaussian processes, it would be desirable to explore similar limitations if the depth is fixed and a highly structured function needs to be learned.

## 4 Proofs

We first start with some preliminary facts concerning GPs and the Gaussian white noise model (2). For $\Pi$ a GP prior on $L^2[0,1]^d$, we can express its covariance kernel $K$ in (3) in terms of the normalized eigenfunctions $\{\phi_k : k \geq 1\}$ of the compact, self-adjoint, trace-class operator $g \mapsto \int_{[0,1]^d} K(\cdot, y)g(y)dy$ on $L^2[0,1]^d$ and its associated eigenvalues $\lambda_k > 0$, which are summable, via the expansion $K(x, y) = \sum_{k=1}^{\infty} \lambda_k \phi_k(x)\phi_k(y)$. In particular, we can realize the Gaussian random element $f \sim \Pi$ through its series expansion

$$f = \sum_{k=1}^{\infty} \sqrt{\lambda_k}\xi_k\phi_k, \qquad \xi_k \sim^{iid} N(0, 1), \qquad (5)$$

known as the *Karhunen-Loève expansion*. The corresponding reproducing kernel Hilbert space (RKHS) $\mathbb{H}$ of $\Pi$ is

$$\mathbb{H} = \left\{ h = \sum_{k=1}^{\infty} h_k \phi_k \; : \; \|h\|_{\mathbb{H}}^2 := \sum_{k=1}^{\infty} \frac{h_k^2}{\lambda_k} < \infty \right\}.$$

For any orthonormal basis $\{\phi_k : k \geq 1\}$ of $L^2[0,1]^d$, in particular the Karhunen-Loève basis of the prior (5), the Gaussian white noise model (2) is equivalent to observing the sequence model $(Y_k : k \geq 1)$:

$$Y_k = \int_{[0,1]^d} \phi_k(x) dY_x = \langle \phi_k, f \rangle_{L^2} + \frac{1}{\sqrt{n}} \int_{[0,1]^d} \phi_k(x) dW_x =: \theta_k + \frac{w_k}{\sqrt{n}}, \tag{6}$$

for $k = 1, 2, 3, \ldots$ and where $w_k \overset{iid}{\sim} N(0, \|\phi_k\|_{L^2}^2) = N(0,1)$. To be precise, the observations $(Y_k : k \geq 1)$ in the last equation and the original white noise model (2) are equivalent in the sense that each can be perfectly recovered from the other, see Section C.2 in the supplement for more details.

Viewing the prior $\Pi$ through its series expansion (5), $\Pi$ can be viewed as a prior on the space of coefficients in the basis expansion of $\{\phi_k\}$ leading to the prior distribution $f = (\theta_k)_k \sim \otimes_{k=1}^{\infty} N(0, \lambda_k)$. Denoting by $P_f = P_f^n$ the distribution of the sequence representation (6), we have likelihood

$$P_f = \otimes_{k=1}^{\infty} N(\theta_k, 1/n).$$

Using this representation, one can use posterior conjugacy to show that the posterior takes the form

$$f = \sum_{k=1}^{\infty} \theta_k \phi_k, \qquad \theta_k | Y_k \sim^{ind} N\left( \frac{n\lambda_k}{n\lambda_k + 1} Y_k, \frac{\lambda_k}{n\lambda_k + 1} \right), \tag{7}$$

see Section C.2 in the supplement for full details. In particular, the posterior mean is

$$\bar{f}_n = E^{\Pi_n}[f|Y] = \sum_{k=1}^{\infty} \frac{n\lambda_k}{n\lambda_k + 1} Y_k \phi_k. \tag{8}$$

## 4.1 Reducing the lower bound for posterior contraction to one for the $L^2$-risk of the posterior mean

We will show that a lower bound on the $L^2$-risk of the posterior mean implies a corresponding lower bound for the posterior contraction rate, so that it will suffice to consider the former. The proofs of the following lemmas are deferred to Section A.1 in the supplement due to space constraints. We start by showing that the squared estimation error of the posterior mean is of the order of its expectation with high probability.

**Lemma 4.** *Let $f_0 \in L^2[0,1]^d$, $\bar{f}_n = E^{\Pi}[f|Y]$ denote the posterior mean based on a mean-zero Gaussian process prior $\Pi$ on $L^2[0,1]^d$ and set $\mu_n^2 = E_{f_0}\|\bar{f}_n - f_0\|_{L^2}^2$. Then for $n \geq 1$,*

$$P_{f_0}\left( \|\bar{f}_n - f_0\|_{L^2}^2 \leq \mu_n^2/4 \right) \leq 4e^{-\frac{n\mu_n^2}{32}}.$$

We next show that a contraction rate for Gaussian process priors implies that the corresponding posterior means converge to the truth at the same rate with frequentist probability tending to one.

**Lemma 5.** *Let $\Pi_n$ be a sequence of mean-zero Gaussian process priors on $L^2[0,1]^d$. Then the corresponding posterior means $\bar{f}_n = E^{\Pi_n}[f|Y]$ satisfy*

$$P_{f_0}\left( \|\bar{f}_n - f_0\|_{L^2} \geq 2\gamma_n \right) \leq 2\sqrt{E_{f_0}\Pi_n(f : \|f - f_0\|_{L^2} \geq \gamma_n | Y)}$$

*for any sequence $\gamma_n$ and $n \geq 1$.*

Combining the last two lemmas allows us to transfer an $L^2$-risk bound for the posterior mean to one for posterior contraction.

**Lemma 6.** *Let $f_0 \in L^2[0,1]^d$, $\Pi_n$ be a sequence of mean-zero Gaussian process priors with corresponding posterior means $\bar{f}_n = E^{\Pi_n}[f|Y]$, and set $\mu_n^2 = E_{f_0}\|\bar{f}_n - f_0\|_{L^2}^2$. Then for $n \geq 1$,*

$$E_{f_0}\Pi_n(f : \|f - f_0\|_{L^2} \geq \mu_n/4|Y) \geq \frac{1}{4}\left(1 - 4e^{-\frac{n\mu_n^2}{32}}\right)_+^2.$$

*Furthermore, suppose $\mathcal{F}_n \subset L^2$ satisfy $\sup_{f_0 \in \mathcal{F}_n} E_{f_0}\|\bar{f}_n - f_0\|_{L^2}^2 \geq \gamma_n^2 > 0$ for some sequence $\gamma_n$ for which $n\gamma_n^2 \to \infty$ as $n \to \infty$. Then for any $0 < \delta < 1/4$ and $n$ such that $n\gamma_n^2 \geq 32\log\left(\frac{5}{1-\sqrt{1-4\delta}}\right)$,*

$$\sup_{f_0 \in \mathcal{F}_n} E_{f_0}\Pi_n(f : \|f - f_0\|_{L^2} \geq \gamma_n/5|Y) \geq 1/4 - \delta.$$

*In particular, the posteriors cannot contract at rate $\gamma_n/5$, uniformly over $\mathcal{F}_n$.*

### 4.2  A general lower bound for the $L^2$-risk of the posterior mean

By Lemma 6, it suffices to prove a lower bound for the $L^2$-risk of the posterior mean. We next prove such a general lower bound.

**Lemma 7.** *Let $\bar{f}_n$ be an estimator of the form (8) with corresponding orthonormal basis $\{\phi_k : k \geq 1\}$. For functions $f_1, \ldots, f_m \in L^2[0,1]^d$,*

$$\max_{j=1,\ldots,m} E_{f_j}\|\bar{f}_n - f_j\|_{L^2}^2 \geq \sum_{k=1}^{\infty}\left(\frac{1}{m}\sum_{j=1}^{m}\langle f_j, \phi_k\rangle_{L^2}^2\right) \wedge \frac{1}{n}. \tag{9}$$

*Moreover, if $f_1, \ldots, f_m$ are orthogonal and $\max_j \|f_j\|_{L^2}^2 \leq m/n$, then*

$$\max_{j=1,\ldots,m} E_{f_j}\|\bar{f}_n - f_j\|_{L^2}^2 \geq \frac{1}{m}\sum_{j=1}^{m}\|f_j\|_{L^2}^2.$$

*In particular, if $cm/n \leq \|f_j\|_{L^2}^2 \leq m/n$ for all $j$ and some $0 < c \leq 1$, then one can take the lower bound to be $cm/n$.*

*Proof.* Using the bias-variance decomposition, the expected prediction loss of an estimator of the form (8) is

$$E_f\|\bar{f}_n - f\|_{L^2}^2 = \sum_{k=1}^{\infty}\left(\frac{n\lambda_k}{1 + n\lambda_k} - 1\right)^2\langle f, \phi_k\rangle_{L^2}^2 + \left(\frac{n\lambda_k}{1 + n\lambda_k}\right)^2\frac{1}{n}.$$

Setting $a_k := n\lambda_k/(1 + n\lambda_k)$ and $T_k = \frac{1}{m}\sum_{j=1}^{m}\langle f_j, \phi_k\rangle_{L^2}^2$, the last equation yields

$$\max_{j=1,\ldots,m} E_{f_j}\|\bar{f}_n - f_j\|_{L^2}^2 \geq \frac{1}{m}\sum_{j=1}^{m} E_{f_j}\|\bar{f}_n - f_j\|_{L^2}^2 = \frac{1}{m}\sum_{j=1}^{m}\sum_{k=1}^{\infty}(a_k - 1)^2\langle f_j, \phi_k\rangle_{L^2}^2 + \frac{a_k^2}{n}$$

$$= \sum_{k=1}^{\infty}(a_k - 1)^2 T_k + \frac{a_k^2}{n}.$$

Optimizing over $a_k$ shows that the right-hand side is minimized at $a_k^* = nT_k/(1 + nT_k)$. For this choice, we obtain using $1 + nT_k \geq 1 \vee (nT_k)$,

$$\max_{j=1,\ldots,m} E_{f_j}\|\bar{f}_n - f_j\|_{L^2}^2 \geq \sum_k \frac{T_k}{(1 + nT_k)^2} + \frac{T_k^2 n}{(1 + nT_k)^2} = \sum_k \frac{T_k}{1 + nT_k} \geq \sum_k T_k \wedge \frac{1}{n},$$

proving the first claim.

To see the second claim, observe that $\|f_j\|_{L^2}^2 \leq m/n$ and the orthogonality of the $f_j$ imply

$$T_k = \frac{1}{m}\sum_{j=1}^{m}\|f_j\|_{L^2}^2\left\langle\frac{f_j}{\|f_j\|_{L^2}}, \phi_k\right\rangle^2 \leq \frac{1}{n}\sum_{j=1}^{m}\left\langle\frac{f_j}{\|f_j\|_{L^2}}, \phi_k\right\rangle^2 \leq \frac{\|\phi_k\|_{L^2}^2}{n} = \frac{1}{n}.$$

Therefore $T_k \wedge 1/n = T_k$ and using (9) and that $(\phi_k)_k$ is a basis,

$$\max_{j=1,\ldots,m} E_{f_j}\|\bar{f}_n - f_j\|_{L^2}^2 \geq \sum_k \frac{1}{m}\sum_{j=1}^{m}\langle f_j, \phi_k\rangle_{L^2}^2 = \frac{1}{m}\sum_{j=1}^{m}\sum_k\langle f_j, \phi_k\rangle_{L^2}^2 = \frac{1}{m}\sum_{j=1}^{m}\|f_j\|_{L^2}^2.$$

$\square$

## 4.3   Construction of local functions and proof of Theorems 1 and 2

**Lemma 8.** *Let $m = k^d$ for some integer $k \geq 1$. Then there exist functions $f_1, \ldots, f_m \in \mathcal{G}$ having disjoint support and satisfying $\|f_j\|_{L^2}^2 = \frac{1}{2(d+2)!} m^{-\frac{2+d}{d}}$ for $j = 1, \ldots, m$.*

*Proof.* For $\ell = 1, \ldots, k$, consider the midpoints $z_\ell := (\ell - 1/2)/k$ of the intervals $[(\ell - 1)/k, \ell/k]$. Overall, there are $m = k^d$ points in the grid

$$I := \left\{ (z_{\ell_1}, \ldots, z_{\ell_d})^\top : \ell_1, \ldots, \ell_d \in \{1, \ldots, k\} \right\}.$$

Consider the functions,

$$f_a(x) = \left( \frac{1}{2k} - |x - a|_1 \right)_+, \qquad a = (a_1, \ldots, a_d) \in I.$$

These functions can be viewed as generalized additive models $f_a(x) = h(\sum_{i=1}^d g_{ai}(x_i))$ with $h(u) = (1/(2k) - u)_+$ and $g_{ai}(x_i) = |x_i - a_i|$. All these functions have Lipschitz constant bounded by 1. Moreover, $\max_{a \in I, i=1,\ldots,d} \|g_{ai}\|_{L^\infty[0,1]} \leq 1$ and $\|h\|_{L^\infty(\mathbb{R})} = 1/(2k) \leq 1$. Thus, $f_a \in \mathcal{G}$.

Since $|x - a|_1 \geq |x - a|_\infty$, the support of the function $f_a$ is contained in the hypercube $\times_{i=1}^d [a_i - 1/(2k), a_i + 1/(2k)]$. Since $a_i$ is of the form $(\ell - 1/2)/k$ with $\ell$ an integer, the functions $f_a$ must have disjoint support and are therefore orthogonal.

It thus remains to compute the $L^2$ norm of these functions, which by translation are the same as that of $f_0(x) = (1/(2k) - |x|_1)_+$ on $\mathbb{R}^d$. Note that the support of $f_0$ can be decomposed as $\operatorname{supp}(f_0) = \cup_{\epsilon \in \{-1,1\}^d} R_\epsilon$, where

$$R_\epsilon = \left\{ x \in \mathbb{R}^d : \operatorname{sign}(x_i) = \epsilon_i, \sum_{j=1}^d |x_j| \leq \frac{1}{2k} \right\}.$$

Since the $L^2$-norm of $f_0$ on each of these $2^d$ quadrants is equal, we have

$$\|f_0\|_{L^2}^2 = 2^d \int_{R_1} f_0(x)^2 dx$$

$$= 2^d \int_{x_1=0}^{1/(2k)} \int_{x_2=0}^{1/(2k)-x_1} \cdots \int_{x_d=0}^{1/(2k)-x_1-\ldots-x_{d-1}} \left( \tfrac{1}{2k} - x_1 - \ldots - x_d \right)^2 dx_1 dx_2 \ldots dx_d.$$

Using repeatedly that $\int_0^{M-r} (M - r - y)^t dy = \frac{1}{t+1}(M - r)^{t+1}$ for $0 \leq r \leq M$ and $t > 0$, we obtain

$$\|f_0\|_{L^2}^2 = 2^d \frac{2}{(d+2)!} \left( \frac{1}{2k} \right)^{d+2} = \frac{1}{2(d+2)!} k^{-d-2}.$$

$\square$

*Proof of Theorems 1 and 2.* Let $r_d = \frac{1}{2(d+2)!}$, $k = \lceil (r_d n)^{\frac{1}{2d+2}} \rceil$ and $m = k^d$. By Lemma 8, there exist orthogonal functions $f_1, \ldots, f_m \in \mathcal{G}$ with $\|f_j\|_{L^2}^2 = r_d m^{-\frac{2+d}{d}}$. For such a choice of $k$ and $m$, $cm/n \leq \|f_j\|_{L^2}^2 \leq m/n$ holds if and only if

$$c^{\frac{1}{2d+2}} \leq \frac{(r_d n)^{\frac{1}{2d+2}}}{\lceil (r_d n)^{\frac{1}{2d+2}} \rceil}$$

after rearranging the inequalities. Now the function $\varphi(x) = x/\lceil x \rceil$ satisfies $\varphi(x) \geq \frac{k}{k+1}$ for all $x \geq k \in \mathbb{N}$, so that the right side is lower bounded by 1/2 for $(r_d n)^{\frac{1}{2d+2}} \geq 1$, or equivalently $n \geq 1/r_d$. For $n \geq 1/r_d$, it therefore holds that $cm/n \leq \|f_j\|_{L^2}^2 \leq m/n$ with $c = (1/2)^{2d+2}$. Applying Lemma 7 thus gives the risk bound

$$\sup_{f_0 \in \mathcal{G}} E_{f_0} \|\bar{f}_n - f_0\|_{L^2}^2 \geq \max_{j=1,\ldots,m} E_{f_j} \|\bar{f}_n - f_j\|_{L^2}^2 \geq \frac{cm}{n} \geq \frac{1}{2^{2d+2}} r_d^{\frac{d}{2d+2}} n^{-\frac{2+d}{2+2d}},$$

which proves Theorem 2.

Set

$$N(d, \delta) = 2(d+2)! 2^{\frac{(2d+2)^2}{d}} \left[ 32 \log \left( \frac{5}{1 - \sqrt{1 - 4\delta}} \right) \right]^{\frac{d+2}{d}},$$

and note that $N(d, \delta) \geq 1/r_d$. Theorem 1 then follows from the second last display and applying Lemma 6 with $\mathcal{F}_n = \{f_1, \ldots, f_m\} \subset \mathcal{G}$ and $\gamma_n^2 = 2^{-(2d+2)} r_d^{\frac{d}{2d+2}} n^{-\frac{2+d}{2+2d}}$. $\qquad\square$

**Acknowledgments.** We are grateful to three anonymous referees for helpful comments that improved the manuscript. The research of MG has been supported by the European Research Council under ERC grant agreement No.834275 (GTBB). The research of JSH has been supported by the NWO/STAR grant 613.009.034b and the NWO Vidi grant VI.Vidi.192.021.

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
