# Supplementary material to "On the inability of Gaussian process regression to optimally learn compositional functions"

**Matteo Giordano**[*]
Department of Statistics
University of Oxford
matteo.giordano@stats.ox.ac.uk

**Kolyan Ray**[*]
Department of Mathematics
Imperial College London
kolyan.ray@imperial.ac.uk

**Johannes Schmidt-Hieber**[*]
Department of Applied Mathematics
University of Twente
a.j.schmidt-hieber@utwente.nl

## Abstract

In this supplement we provide additional background material and all the technical results and proofs not included in the main article. For the convenience of the reader, we repeat the statements of results that we prove here. We continue the equation numbering scheme from the main document to the supplement.

## Contents

---

[*]All authors contributed equally.

36th Conference on Neural Information Processing Systems (NeurIPS 2022).

# A  Remaining proofs: Lemmas 4-6, and Theorem 3

## A.1  Proof of Lemmas 4-6

**Lemma 4.** *Let $f_0 \in L^2[0,1]^d$, $\bar{f}_n = E^\Pi[f|Y]$ denote the posterior mean based on a mean-zero Gaussian process prior $\Pi$ on $L^2[0,1]^d$ and set $\mu_n^2 = E_{f_0} \|\bar{f}_n - f_0\|_{L^2}^2$. Then for $n \geq 1$,*

$$P_{f_0}\left( \|\bar{f}_n - f_0\|_{L^2}^2 \leq \mu_n^2/4 \right) \leq 4e^{-\frac{n\mu_n^2}{32}}.$$

*Proof.* Using (8), under $P_{f_0}$ the posterior mean equals

$$\bar{f}_n = \sum_k \frac{n\lambda_k}{n\lambda_k + 1} Y_k \phi_k = \sum_k \frac{n\lambda_k}{n\lambda_k + 1} f_{0k}\phi_k + \sum_k \frac{n\lambda_k}{n\lambda_k + 1} \frac{w_k}{\sqrt{n}} \phi_k.$$

Since $f_0 = \sum_k f_{0k}\phi_k$, we deduce

$$
\begin{aligned}
\|\bar{f}_n - f_0\|_{L^2}^2 &= \sum_k \left( \frac{\sqrt{n}\lambda_k}{n\lambda_k + 1} w_k - \frac{1}{n\lambda_k + 1} f_{0k} \right)^2 \\
&= \sum_k \frac{1}{(n\lambda_k + 1)^2} \left[ n\lambda_k^2(w_k^2 - 1) + n\lambda_k^2 - 2\sqrt{n}\lambda_k w_k f_{0k} + f_{0k}^2 \right] \\
&= I + II + III + IV.
\end{aligned}
$$

Taking $E_{f_0}$-expectation yields

$$E_{f_0}\|\bar{f}_n - f_0\|_{L^2}^2 = \sum_k \frac{n\lambda_k^2}{(n\lambda_k + 1)^2} + \frac{f_{0k}^2}{(n\lambda_k + 1)^2} = II + IV.$$

We will now show that $I + III$ are of smaller order with high $P_{f_0}$-probability, so that $\|\bar{f}_n - f_0\|_{L^2}^2$ is close to its expectation, and hence the size of its expectation drives its behaviour.

*I*: Using Lemma 1 of [4] with $a_k = \frac{n\lambda_k^2}{(n\lambda_k+1)^2}$, we get

$$P\left( |I| \geq 2\|a\|_{\ell_2}\sqrt{x} + 2\|a\|_{\ell_\infty} x \right) \leq 2e^{-x}$$

for any $x > 0$ (we write $P$ instead of $P_{f_0}$ here to emphasize the above probability does not depend on $f_0$). Further set $\alpha_k = \frac{\lambda_k}{\lambda_k + 1/n}$, so that $na_k = \alpha_k^2$. Then

$$n^2\|a\|_{\ell_2}^2 = \|\alpha\|_{\ell_4}^4 \leq \sup_k |\alpha_k|^2 \sum_k \alpha_k^2 = \sup_k \left| \frac{\lambda_k^2}{(\lambda_k + 1/n)^2} \right| \|\alpha\|_{\ell_2}^2 \leq \|\alpha\|_{\ell_2}^2.$$

Using that $\|\alpha\|_{\ell_2}^2 = nII$, this implies $\|a\|_{\ell_2} \leq \frac{1}{n}\|\alpha\|_{\ell_2} = \sqrt{II/n}$. For the $\ell_\infty$ term,

$$\|a\|_{\ell_\infty} = \frac{1}{n}\sup_k \left| \frac{n^2\lambda_k^2}{(n\lambda_k+1)^2} \right| \leq \frac{1}{n}.$$

In conclusion, we have the exponential inequality

$$P\left( |I| \geq 2\sqrt{\frac{IIx}{n}} + \frac{2x}{n} \right) \leq 2e^{-x}.$$

*III*: We have $III \sim N(0, \sum_k \frac{4n\lambda_k^2 f_{0k}^2}{(n\lambda_k+1)^4})$, whose variance is bounded by $\frac{4}{n}\sup_k \frac{n^2\lambda_k^2}{(n\lambda_k+1)^2} \sum_k \frac{f_{0k}^2}{(n\lambda_k+1)^2} \leq \frac{4}{n}IV$. Thus using the standard Gaussian tail bound, for any $x \geq 0$,

$$P_{f_0}\left( |III| \geq 2\sqrt{IV/n}x \right) \leq 2e^{-x^2/2},$$

uniformly over $f_0 \in L^2$.

The triangle inequality gives $\left|\|\bar{f}_n - f_0\|_{L^2}^2 - E_{f_0}\|\bar{f}_n - f_0\|_{L^2}^2\right| = |I + III| \leq |I| + |III|$. Since for any real numbers $a, b$, $P_{f_0}(|I| + |III| \geq a + b) \leq P_{f_0}(\{|I| \geq a\} \cup \{|III| \geq b\}) \leq P_{f_0}(|I| \geq a) + P_{f_0}(|III| \geq b)$, combining the bounds for $I$ and $III$, we get

$$\sup_{f_0 \in L^2} P_{f_0}\left(\left|\|\bar{f}_n - f_0\|_{L^2}^2 - E_{f_0}\|\bar{f}_n - f_0\|_{L^2}^2\right| \geq 2\sqrt{\frac{IIx}{n}} + \frac{2x}{n} + \sqrt{\frac{8IVx}{n}}\right) \leq 4e^{-x}$$

for all $x > 0$, a non-asymptotic inequality. Using $ab \leq (a^2 + b^2)/2$ gives $\sqrt{8IVx/n} \leq IV/2 + 4x/n$ and $2\sqrt{IIx/n} \leq II/2 + 2x/n$. Since $E_{f_0}\|\bar{f}_n - f_0\|_{L^2}^2 = II + IV$, it holds that for any $x > 0$,

$$\sup_{f_0 \in L^2} P_{f_0}\left(\left|\|\bar{f}_n - f_0\|_{L^2}^2 - E_{f_0}\|\bar{f}_n - f_0\|_{L^2}^2\right| \geq \frac{1}{2}E_{f_0}\|\bar{f}_n - f_0\|_{L^2}^2 + \frac{8x}{n}\right) \leq 4e^{-x}.$$

Set $x = n\mu_n^2/32$ so that $8x/n = \mu_n^2/4$. Then the last inequality implies that for all $f_0 \in L^2$,

$$P_{f_0}\left(\|\bar{f}_n - f_0\|_{L^2}^2 \leq \frac{1}{2}E_{f_0}\|\bar{f}_n - f_0\|_{L^2}^2 - \frac{\mu_n^2}{4}\right) \leq 4e^{-\frac{n\mu_n^2}{32}}.$$

Substituting in $\mu_n^2 = E_{f_0}\|\bar{f}_n - f_0\|_{L^2}^2$ gives the result. $\qquad\square$

**Lemma 5.** *Let $\Pi_n$ be a sequence of mean-zero Gaussian process priors on $L^2[0,1]^d$. Then the corresponding posterior means $\bar{f}_n = E^{\Pi_n}[f|Y]$ satisfy*

$$P_{f_0}\left(\|\bar{f}_n - f_0\|_{L^2} \geq 2\gamma_n\right) \leq 2\sqrt{E_{f_0}\Pi_n(f : \|f - f_0\|_{L^2} \geq \gamma_n|Y)}$$

*for any sequence $\gamma_n$ and $n \geq 1$.*

*Proof.* Write $v_n = E_{f_0}\Pi_n(f : \|f - f_0\|_{L^2} \geq \gamma_n|Y)$. We may assume $v_n < 1/4$, otherwise the right side is greater than one and there is nothing to prove. Using Markov's inequality, the events $A_{n,f_0} = \{\Pi_n(f : \|f - f_0\|_{L^2} \geq \gamma_n|Y) \leq \sqrt{v_n}\}$ then satisfy $P_{f_0}(A_{n,f_0}^c) \leq \sqrt{v_n}$.

Recall that the posterior distributions $\Pi_n(f|Y)$ are also Gaussian by conjugacy, see (7). Since $\{f \in L^2 : \|f\|_{L^2} \leq \gamma_n\}$ is convex and symmetric, Anderson's inequality (e.g. [3], Theorem 2.4.5) implies that

$$1 - v_n = E_{f_0}\Pi_n(f : \|f - f_0\|_{L^2} \leq \gamma_n|Y) \leq E_{f_0}\Pi_n(f : \|f - \bar{f}_n\|_{L^2} \leq \gamma_n|Y).$$

Using the same argument as above, we thus have $P_{f_0}(B_n^c) \leq \sqrt{v_n}$ for $B_n = \{\Pi_n(f : \|f - \bar{f}_n\|_{L^2} \geq \gamma_n|Y) \leq \sqrt{v_n}\}$, so that $P_{f_0}(A_{n,f_0} \cap B_n) \geq 1 - 2\sqrt{v_n}$. Now on the event $A_{n,f_0} \cap B_n$, we have

$$\Pi_n(f : \|f - f_0\|_{L^2} \leq \gamma_n, \|f - \bar{f}_n\|_{L^2} \leq \gamma_n|Y) \geq 1 - \Pi_n(f : \|f - f_0\|_{L^2} \geq \gamma_n|Y)$$
$$- \Pi_n(f : \|f - \bar{f}_n\|_{L^2} \geq \gamma_n|Y)$$
$$\geq 1 - 2\sqrt{v_n}.$$

The right-hand side is strictly positive for $v_n < 1/4$ and hence so is the left posterior probability. Thus there must exist $f \in L^2$ in the left set, in which case

$$\|\bar{f}_n - f_0\|_{L^2} \leq \|\bar{f}_n - f\|_{L^2} + \|f - f_0\|_{L^2} \leq 2\gamma_n.$$

We have thus shown that $P_{f_0}(\|\bar{f}_n - f_0\|_{L^2} \leq 2\gamma_n) \geq 1 - 2\sqrt{v_n}$ as required. $\qquad\square$

**Lemma 6.** *Let $f_0 \in L^2[0,1]^d$, $\Pi_n$ be a sequence of mean-zero Gaussian process priors with corresponding posterior means $\bar{f}_n = E^{\Pi_n}[f|Y]$, and set $\mu_n^2 = E_{f_0}\|\bar{f}_n - f_0\|_{L^2}^2$. Then for $n \geq 1$,*

$$E_{f_0}\Pi_n(f : \|f - f_0\|_{L^2} \geq \mu_n/4|Y) \geq \frac{1}{4}\left(1 - 4e^{-\frac{n\mu_n^2}{32}}\right)_+^2.$$

*Furthermore, suppose $\mathcal{F}_n \subset L^2$ satisfy $\sup_{f_0 \in \mathcal{F}_n} E_{f_0}\|\bar{f}_n - f_0\|_{L^2}^2 \geq \gamma_n^2 > 0$ for some sequence $\gamma_n$ for which $n\gamma_n^2 \to \infty$ as $n \to \infty$. Then for any $0 < \delta < 1/4$ and $n$ such that $n\gamma_n^2 \geq 32\log\left(\frac{5}{1-\sqrt{1-4\delta}}\right)$,*

$$\sup_{f_0 \in \mathcal{F}_n} E_{f_0}\Pi_n(f : \|f - f_0\|_{L^2} \geq \gamma_n/5|Y) \geq 1/4 - \delta.$$

*In particular, the posteriors cannot contract at rate $\gamma_n/5$, uniformly over $\mathcal{F}_n$.*

*Proof.* Using Lemma 5 and then Lemma 4, the square-root of the left-side of the first display is lower bounded by

$$\frac{1}{2} P_{f_0} \left( \|\bar{f}_n - f_0\|_{L^2} \geq \mu_n/2 \right) \geq \frac{1}{2} \left( 1 - 4e^{-\frac{n\mu_n^2}{32}} \right).$$

Squaring everything then gives the first result.

By hypothesis, for any $\eta > 0$, there exists $f_{0,n} \in \mathcal{F}_n$ such that $\mu_n^2 = E_{f_{0,n}} \|\bar{f}_n - f_{0,n}\|_{L^2}^2 \geq \gamma_n^2 - \eta$. Let $\eta = \eta_n$ be small enough that $e^{\frac{n\eta}{32}} \leq 5/4$ and $\sqrt{\gamma_n^2 - \eta}/4 \geq \gamma_n/5$. By the first part of the lemma,

$$E_{f_{0,n}} \Pi_n (f : \|f - f_{0,n}\|_{L^2} \geq \sqrt{\gamma_n^2 - \eta}/4 | Y) \geq \frac{1}{4} \left( 1 - 5e^{-\frac{n\gamma_n^2}{32}} \right)_+^2.$$

After rearranging, the right-side is at least $1/4 - \delta$ for $n\gamma_n^2 \geq 32 \log \left( \frac{5}{1 - \sqrt{1 - 4\delta}} \right)$. The second part then follows from upper bounding the left-side of the last display by $E_{f_{0,n}} \Pi_n (f : \|f - f_{0,n}\|_{L^2} \geq \gamma_n/5 | Y)$. $\qquad \square$

## A.2  Proof of Theorem 3

Similar to the proof of Theorems 1 and 2, we derive a lower bound for the $L^2$-risk of the posterior mean over a suitable subset of the class $\mathcal{G}$.

For a wavelet prior $\Pi_n$ defined as in (4) (with possibly $n$-dependent $\lambda_\gamma$) and $\bar{f}_n$ the corresponding posterior mean, the conclusion of Lemma 7 reads, for $m = 1$,

$$E_f \|\bar{f}_n - f\|_{L^2}^2 \geq \sum_{\gamma \in \Gamma^d} \langle f, \psi_\gamma \rangle_{L^2}^2 \wedge \frac{1}{n}, \qquad \text{any } f \in L^2[0,1]^d. \tag{10}$$

By applying Lemma 2 in [5] (with $\alpha = K = 1$), for a constant $c(d, \psi) > 0$ only depending on $d$ and the wavelet basis, letting $j_n \in \mathbb{N}$ satisfy

$$\frac{1}{n} \leq c(d, \psi) 2^{-j_n(2+d)} \leq \frac{2^{2+d}}{n}$$

there exists for each $n$ a Lipschitz function $h_{j_n} : [0, d] \to [0, 1]$ with Lipschitz constant bounded by 1 such that the generalized additive function $f_{j_n}(x_1, \dots, x_d) := h_{j_n}(x_1 + \dots + x_d)$ satisfies for all $p_1, \dots, p_d \in \{0, 1, \dots, 2^{j_n - q - \nu} - 1\}$,

$$\langle f_{j_n}, \psi_{(j_n, 2^{q+\nu} p_1), \dots, (j_n, 2^{q+\nu} p_d)} \rangle_{L^2}^2 = c(d, \psi)^2 2^{-j_n(2+d)} \geq \frac{1}{n}.$$

Above, $q > 0$ is such that the mother wavelet $\psi$ is supported within $[0, 2^q]$ and $\nu = \lceil \log_2 d \rceil + 1$. Noting that with our choice of $j_n$ we have $2^{j_n} \geq \frac{1}{2}(c(d, \psi)^2 n)^{\frac{1}{2+d}}$, it follows from (10) that

$$E_{f_{j_n}} \|\bar{f}_n - f_{j_n}\|_{L^2}^2 \geq \sum_{p_1, \dots, p_d \in \{0, 1, \dots, 2^{j_n - q - \nu} - 1\}} \frac{1}{n}$$

$$= 2^{-(qd + \nu d)} \frac{1}{n} 2^{j_n d} \geq 2^{-(q + \nu + 1)d} (c(d, \psi))^{\frac{2d}{2+d}} n^{-\frac{2}{2+d}}.$$

Since $f_{j_n} \in \mathcal{G}$, this proves the second claim of Theorem 3. Another application of Lemma 6 with $\mathcal{F}_n = \{f_{j_n}\} \subset \mathcal{G}$, $\gamma_n^2 = 2^{-(q+\nu+1)d}(c(d, \psi))^{\frac{2d}{2+d}} n^{-\frac{2}{2+d}}$ then proves the first claim taking $N(d, \delta) \geq 32 \log \left( \frac{5}{1 - \sqrt{1 - 4\delta}} \right) / \gamma_n^2$ and $C = C(d, \psi) = 2^{-(q+\nu+1)d/2}(c(d, \psi))^{\frac{d}{2+d}}/5$.

## B  An alternative proof of Theorems 1 and 2

As part of the proofs of Theorems 1 and 2 in Section 4, we directly lower bounded the $L^2$-risk of the posterior mean in Lemma 7. We now provide an alternative strategy to prove a lower bound via first reducing the regression setting to a one-sparse sequence model in which we can explicitly evaluate the minimax risk for linear estimators.

## B.1 Reduction to a one-sparse sequence model

Consider the Gaussian white noise model (2) with finite parameter space $\mathcal{F}_n = \{f_1, \ldots, f_m\}$, $m = m_n$, satisfying

$$\langle f_i, f_j \rangle_{L^2} = \delta_{ij} c_n^2, \qquad c_n > 0 \tag{11}$$

[we will ultimately take $f_1, \ldots, f_m \in \mathcal{G}$ as in Lemma 8, but this approach may be useful for other function classes]. Denote by $f \in \mathcal{F}_n$ the regression function driving (2) and define

$$y_i := \frac{1}{c_n^2} \int_{[0,1]^d} f_i(x) dY_x = \frac{1}{c_n^2} \langle f_i, f \rangle_{L^2} + \frac{1}{\sqrt{n} c_n^2} \int_{[0,1]^d} f_i(x) dW_x, \tag{12}$$

$i = 1, \ldots, m$. Further write

$$\theta_i := \frac{1}{c_n^2} \langle f_i, f \rangle_{L^2}, \qquad w_i := \frac{1}{c_n} \int_{[0,1]^d} f_i(x) dW_x \sim^{iid} N(0,1), \qquad \sigma_n := \frac{1}{c_n \sqrt{n}}. \tag{13}$$

Since $f \in \mathcal{F}_n$ in our statistical model satisfy (11), $\theta = (\theta_1, \ldots, \theta_m)^T$ is a *one-sparse* vector. The present statistical model thus yields an observation from the finite Gaussian sequence model

$$y_i = \theta_i + \sigma_n w_i, \qquad i = 1, \ldots, m, \tag{14}$$

with $\theta \in \Theta_m = \{e_1, \ldots, e_m\}$ for $e_i$ the $i^{th}$ basis vector, that is $e_{ij} = \delta_{ij}$.

We now relate estimation in the full Gaussian white noise model with $f \in \mathcal{F}_n$ with estimation in (14). Since $\theta_i = \langle f_i, f \rangle_{L^2}/c_n^2$, $f_i/c_n$ are $L^2$-normalized and orthogonal, and $f \in \mathcal{F}_n$ trivially lies in the linear span of $\{f_1, \ldots, f_m\}$, we can express the function $f$ in terms of $\theta = (\theta_1, \ldots, \theta_m)^T$ via

$$f = \sum_{i=1}^m \frac{\langle f_i, f \rangle_{L^2}}{c_n} \frac{f_i}{c_n} = \sum_{i=1}^m \theta_i f_i.$$

Any estimator $\hat{\theta} = (\hat{\theta}_1, \ldots, \hat{\theta}_m)^T$ for $\theta$ thus yields a series estimator $\hat{f}_{\hat{\theta}} = \sum_{i=1}^m \hat{\theta}_i f_i$ for $f \in \mathcal{F}_n$, and

$$\|\hat{f}_{\hat{\theta}} - f\|_{L^2}^2 = \sum_{i=1}^m (\hat{\theta}_i - \theta_i)^2 \|f_i\|_{L^2}^2 = c_n^2 \sum_{i=1}^m (\hat{\theta}_i - \theta_i)^2 = c_n^2 |\hat{\theta} - \theta|^2. \tag{15}$$

We next show that in the full Gaussian white noise model with restricted parameter space $\mathcal{F}_n$, the risk of the posterior mean is larger than that of an estimator of the form $\hat{f}_{\hat{\theta}}$ with $\hat{\theta} = Ay$ a linear function of the observations $y = (y_1, \ldots, y_m)^T$ in (12).

**Lemma 7.** *Let $\Pi_n$ be a sequence of mean-zero Gaussian process priors on $L^2[0,1]^d$ with corresponding posterior means $\bar{f}_n = E^{\Pi_n}[f|Y]$. Then there exists a matrix sequence $A_n \in \mathbb{R}^{m \times m}$ such that for every $n$ and every $f \in \mathcal{F}_n = \{f_1, \ldots, f_m\}$ satisfying (11),*

$$E_f \|\bar{f}_n - f\|_{L^2} \geq E_f \|\hat{f}_{A_n} - f\|_{L^2},$$

*where $\hat{f}_{A_n} = \sum_{i=1}^m (A_n y)_i f_i$ and $y_1, \ldots, y_m$ are defined in (12).*

*Proof.* Recall from (8) that the posterior mean takes the form $\bar{f}_n = \sum_{k=1}^{\infty} a_k Y_k \phi_k$, where $a_k = \frac{n\lambda_k}{1+n\lambda_k}$ with $(\lambda_k)$, $(\phi_k)$ the eigenvalues/vectors arising in the Karhunen-Loève expansion (5), and $Y_k = \langle Y, \phi_k \rangle_{L^2}$ defined in (6). We can decompose each $\phi_k$ as

$$\phi_k = \psi_k + g_k, \qquad \psi_k = \sum_{i=1}^m \psi_{k,i} f_i \in \text{span}\{f_1, \ldots, f_m\}, \qquad g_k \perp \{f_1, \ldots, f_m\}$$

using the orthogonality relation (11), giving

$$\bar{f}_n = \sum_{k=1}^{\infty} a_k \langle Y, \psi_k + g_k \rangle_{L^2} \psi_k + \sum_{k=1}^{\infty} a_k \langle Y, \psi_k + g_k \rangle_{L^2} g_k =: \bar{f}_1 + \bar{f}_2.$$

Define the matrix $A_n \in \mathbb{R}^{m \times m}$ by

$$(A_n)_{ij} = c_n^2 \sum_{k=1}^{\infty} a_k \psi_{k,i} \psi_{k,j}.$$

Using that $\langle Y, f_j \rangle_{L^2} = c_n^2 y_j$ from (12), we can rewrite

$$\bar{f}_1 = \sum_{k=1}^{\infty} a_k \left( \sum_{j=1}^{m} \psi_{k,j} \langle Y, f_j \rangle_{L^2} + \langle Y, g_k \rangle_{L^2} \right) \sum_{i=1}^{m} \psi_{k,i} f_i$$

$$= \sum_{i=1}^{m} \left( \sum_{j=1}^{m} \left[ \sum_{k=1}^{\infty} a_k \psi_{k,i} \psi_{k,j} \right] c_n^2 y_j + \sum_{k=1}^{\infty} a_k \psi_{k,i} \langle Y, g \rangle_{L^2} \right) f_i$$

$$= \sum_{i=1}^{m} \left( (A_n y)_i + \sum_{k=1}^{\infty} a_k \psi_{k,i} \langle Y, g \rangle_{L^2} \right) f_i.$$

Since $\bar{f}_2 \perp f$ for all $f \in \mathcal{F}_n$, $\|f - \bar{f}_n\|_{L^2}^2 = \|f - \bar{f}_1\|_{L^2}^2 + \|\bar{f}_2\|_{L^2}^2 \geq \|f - \bar{f}_1\|_{L^2}^2$. Writing $f = \sum_{i=1}^{m} \theta_i f_i$, where $\theta_i = \langle f, f_i \rangle_{L^2}/c_n^2$, and recalling $\langle f_i, f_j \rangle_{L^2} = \delta_{ij} c_n^2$,

$$\|f - \bar{f}_1\|_{L^2}^2 = \left\| \sum_{i=1}^{m} \left( \theta_i - (A_n y)_i - \sum_{k=1}^{\infty} a_k \psi_{k,i} \langle Y, g_k \rangle_{L^2} \right) f_i \right\|_{L^2}^2$$

$$= c_n^2 \sum_{i=1}^{m} \left( \theta_i - (A_n y)_i - \sum_{k=1}^{\infty} a_k \psi_{k,i} \langle Y, g_k \rangle_{L^2} \right)^2.$$

Using the independence between $y_i = \langle Y, f_i \rangle_{L^2}/c_n^2$ and $\langle Y, g_k \rangle_{L^2}$ for all $i = 1, \ldots, m$ and $k \geq 1$, and that $E_f \langle Y, g_k \rangle_{L^2} = 0$, we obtain that for all $f \in \mathcal{F}_n$,

$$E_f \|f - \bar{f}_n\|_{L^2}^2 \geq c_n^2 E_f \sum_{i=1}^{m} \left\{ (\theta_i - (A_n y)_i)^2 + \left( \sum_{k=1}^{\infty} a_k \psi_{k,i} \langle Y, g_k \rangle_{L^2} \right)^2 \right\}$$

$$\geq c_n^2 \sum_{i=1}^{m} E_f \left[ \theta_i - (A_n y)_i \right]^2 = E_f \|f - \hat{f}_{A_n}\|_{L^2}^2.$$

$\square$

Using Lemma 7 and (15), we thus obtain

$$\inf_{\bar{f}_n} \max_{f \in \mathcal{F}_n} E_f \|\bar{f}_n - f\|_{L^2}^2 \geq c_n^2 \inf_{A \in \mathbb{R}^{m \times m}} \max_{\theta \in \Theta_m} |Ay - \theta|^2, \tag{16}$$

where the infimum on the left side is over all posterior means based on mean-zero Gaussian processes. If thus suffices to lower bound the right-side, which is the minimax risk for *linear* estimators in the one-sparse sequence model (14).

## B.2 Minimax risk for linear estimators in the one-sparse Gaussian sequence model

We now study the minimax risk for linear estimators in the one-sparse model (14) for arbitrary noise level $\sigma_n > 0$. Recall that $\theta = (\theta_1, \ldots, \theta_m)^T$ has exactly one non-zero coordinate, which is equal to one, so that the parameter space is $\Theta_m = \{e_1, \ldots, e_m\}$ for $e_i$ the $i^{th}$ basis vector, i.e. $e_{ij} = \delta_{ij}$.

A *linear* estimator of $\theta$ in model (14) takes the form

$$\hat{\theta}_A = Ay,$$

for some matrix $A \in \mathbb{R}^{m \times m}$ and $y = (y_1, \ldots, y_m)^T$. We now show that in this one-sparse model, such estimators are dominated by *diagonal homogeneous* linear estimators in terms of their maximal mean-squared error $E_\theta |\hat{\theta}_A - \theta|^2$, where $|\cdot|$ denotes the usual Euclidean norm on $\mathbb{R}^m$ and $E_\theta$ the expectation in model (14) with true parameter $\theta$. Hence the minimax risk for linear estimators is attained by a diagonal homogeneous linear estimator.

**Lemma 8.** *Let $A = (a_{ij}) \in \mathbb{R}^{m \times m}$ be any matrix and let $\bar{a}^2 = \frac{1}{m} \sum_{j=1}^{m} a_{jj}^2$. Then*

$$\max_{\theta \in \Theta_m} E_\theta |\hat{\theta}_A - \theta|^2 \geq \max_{\theta \in \Theta_m} E_\theta |\hat{\theta}_{\bar{a} I_m} - \theta|^2.$$

*In particular,*

$$\inf_{A \in \mathbb{R}^{m \times m}} \max_{\theta \in \Theta_m} E_\theta |\hat{\theta}_A - \theta|^2 = \inf_{a \in \mathbb{R}} \max_{\theta \in \Theta_m} E_\theta |\hat{\theta}_{a I_m} - \theta|^2.$$

*Proof.* Using the bias-variance decomposition,

$$
\begin{aligned}
E_\theta |\hat\theta_A - \theta|^2 &= |E_\theta \hat\theta_A - \theta|^2 + \text{tr}[\text{Cov}_\theta(\hat\theta_A)] \\
&= |(A - I_m)\theta|^2 + \text{tr}[A\text{Cov}_\theta(y)A^T] \\
&= |(A - I_m)\theta|^2 + \sigma_n^2 \text{tr}[AA^T] \\
&= \sum_{i=1}^m \left( \sum_{j=1}^m (a_{ij} - \delta_{ij})\theta_j \right)^2 + \sigma_n^2 \sum_{i,j=1}^m a_{ij}^2,
\end{aligned}
$$

since $\text{Cov}_\theta(y) = \text{Cov}_\theta(w) = \sigma_n^2 I_m$. Since $\theta \in \Theta_m = \{e_1, \ldots, e_m\}$, let $j^* \in \{1, \ldots, m\}$ be the index such that $\theta = e_{j^*}$. Then

$$
E_\theta |\hat\theta_A - \theta|^2 = \sum_{i=1}^m (a_{ij^*} - \delta_{ij^*})^2 + \sigma_n^2 \sum_{i,j=1}^m a_{ij}^2.
$$

However, applying this last expression also with $\tilde A = \text{diag}(A)$, the diagonal matrix with entries $\tilde a_{ij} = a_{ij}\delta_{ij}$, gives

$$
E_\theta |\hat\theta_{\tilde A} - \theta|^2 = (a_{j^* j^*} - 1)^2 + \sigma_n^2 \sum_{i=1}^m a_{ii}^2 \le E_\theta |\hat\theta_A - \theta|^2,
$$

a bound which holds for all $\theta \in \Theta_m$. Thus we need only consider the estimator $\hat\theta_{\tilde A}$ with diagonal matrix $\tilde A$. Using the last display,

$$
\max_{\theta \in \Theta_m} E_\theta |\hat\theta_{\tilde A} - \theta|^2 = \max_{j=1,\ldots,m} (a_{jj} - 1)^2 + \sigma_n^2 m \bar a^2.
$$

Since the matrix $\bar a I_m$ is also diagonal, this further yields

$$
\max_{\theta \in \Theta_m} E_\theta |\hat\theta_{\bar a I_m} - \theta|^2 = (\bar a - 1)^2 + \sigma_n^2 m \bar a^2,
$$

so that it is enough to show $(\bar a - 1)^2 \le \max_j (a_{jj} - 1)^2$. Since the function $\varphi(x) = (\sqrt{x} - 1)^2$ is convex on $(0, \infty)$, Jensen's inequality implies

$$
(\bar a - 1)^2 = \left( \sqrt{\frac{1}{m} \sum_{i=1}^m a_{ii}^2} - 1 \right)^2 \le \frac{1}{m} \sum_{i=1}^m (|a_{ii}| - 1)^2 \le \max_{j=1,\ldots,m} (a_{jj} - 1)^2
$$

as desired. $\qquad\square$

The last lemma immediately gives the minimax risk for linear estimators in this model.

**Lemma 9** (Linear minimax risk in the one-sparse model). *Consider model* (14). *For $\hat\theta_A = Ay$, we have*

$$
\inf_{A \in \mathbb{R}^{m \times m}} \max_{\theta \in \Theta_m} E_\theta |\hat\theta_A - \theta|^2 = \frac{m\sigma_n^2}{1 + m\sigma_n^2}.
$$

*Proof.* Lemma 8 implies that a linear estimator of $\theta$ with minimal maximal risk over $\Theta_m$ necessarily has the form $\hat\theta_{a I_m} = ay$ for some $a \in \mathbb{R}$. For such an estimator and any $\theta \in \Theta_m$, the bias-variance decomposition gives

$$
E_\theta |\hat\theta_{a I_m} - \theta|^2 = (a - 1)^2 + m\sigma_n^2 a^2,
$$

which can be explicitly minimized at $a^* = \frac{1}{1 + m\sigma_n^2}$ with corresponding minimal risk

$$
E_\theta |\hat\theta_{a^* I_m} - \theta|^2 = \frac{(m\sigma_n^2)^2}{(1 + m\sigma_n^2)^2} + \frac{m\sigma_n^2}{(1 + m\sigma_n^2)^2} = \frac{m\sigma_n^2}{1 + m\sigma_n^2}.
$$

$\qquad\square$

### B.3 Proof of Theorems 1 and 2

We do not keep explicit track of constants in this version of the proof, noting simply they will finally depend only on the dimension $d$.

*Proof of Theorems 1 and 2.* Let $r_d = \frac{1}{2(d+2)!}$, $k = \lceil (r_d n)^{\frac{1}{2d+2}} \rceil$ and $m = k^d$. By Lemma 8, there exist orthogonal function $f_1, \ldots, f_m \in \mathcal{G}$ with $\|f_j\|_{L^2}^2 = r_d m^{-\frac{2+d}{d}}$. Set $\mathcal{F}_n = \{f_1, \ldots, f_n\}$ so that (11) is satisfied with $c_n^2 = r_d m^{-\frac{2+d}{d}}$. Using (16) and Lemma 9,

$$\sup_{f_0 \in \mathcal{G}} E_f \|\bar{f}_n - f\|_{L^2}^2 \geq \max_{f \in \mathcal{F}_n} E_f \|\bar{f}_n - f\|_{L^2}^2 \geq c_n^2 \frac{m\sigma_n^2}{1 + m\sigma_n^2} = \frac{m}{n} \frac{1}{1 + m/(nc_n^2)}$$

since $\sigma_n^2 = \frac{1}{nc_n^2}$ by (13). Since $m = m_n$ satisfies $\frac{m}{c_n^2 n} \simeq n^{-1+1/d} \lesssim 1$, we deduce

$$\max_{f \in \mathcal{F}_n} E_f \|\bar{f}_n - f\|_{L^2}^2 \gtrsim \frac{m}{n} \simeq n^{-\frac{2+d}{2+2d}},$$

which proves Theorem 2. Theorem 1 then follows by applying Lemma 6 with $\mathcal{F}_n = \{f_1, \ldots, f_m\} \subset \mathcal{G}$ and $\gamma_n^2 \simeq n^{-\frac{2+d}{2+2d}}$. □

## C Background material

### C.1 Minimax estimation

Describing the large sample behavior of the smallest achievable worst case risk over all estimation procedures, minimax (estimation) rates are a standard tool to establish statistical optimality of a method. For a statistical model $(P_\theta^n : \theta \in \Theta)$ with sample size $n$ and a loss function $\ell$, the minimax risk is $R_n = \inf_{\widehat{\theta}_n} \sup_{\theta \in \Theta} E_\theta[\ell(\widehat{\theta}_n, \theta)]$, where the infimum is taken over all estimators $\widehat{\theta}_n$. The minimax rate is any sequence $(r_n)_n$ such that $r_n \asymp R_n$. Any estimator $\widetilde{\theta}_n$ with $R_n \asymp \sup_{\theta \in \Theta} E_\theta[\ell(\widetilde{\theta}_n, \theta)]$ is called minimax rate optimal. By definition, the risk of minimax rate optimal estimators is at most a constant factor larger than the minimax risk $R_n$.

If the posterior contraction rate is $\varepsilon_n$, then under very weak assumptions one can find an estimator with worst case risk of the order $\varepsilon_n$, see Theorem 2.5 [2]. This in turn implies that the posterior cannot contract faster than the minimax rate.

### C.2 Sequence representation and posterior mean

We provide here some explanation behind the sequence representation (6) of the Gaussian white noise model (2) and the derivation of the posterior distribution (7). Recall that we can realize a random element $f \sim \Pi$ distributed according to a GP prior on $L^2[0,1]^d$ via its series expansion (5), namely

$$f = \sum_{k=1}^{\infty} \sqrt{\lambda_k} \xi_k \phi_k, \qquad \xi_k \sim^{iid} N(0,1),$$

known as the *Karhunen-Loève expansion*.

The Brownian motion $W$ in (2) can equivalently be viewed via the action of integration on test functions $g \in L^2[0,1]^d$ through $W_g = \int_{[0,1]^d} g(x)dW_x$, leading to the mean-zero Gaussian process $W = (W_g : g \in L^2[0,1]^d)$ indexed by $L^2[0,1]^d$ with covariance $E(W_g W_h) = \langle g, h \rangle_{L^2}$. In this form, the Gaussian white noise model (2) can be interpreted as observing the Gaussian process $Y = (Y_g : g \in L^2[0,1]^d)$ with

$$Y_g = \langle f, g \rangle_{L^2} + n^{-1/2} W_g, \qquad g \in L^2[0,1]^d.$$

It is then statistically equivalent to observe the subprocess $(Y_k = Y_{\phi_k} : k \geq 1)$ for any orthonormal basis $\{\phi_k : k \geq 1\}$ of $L^2[0,1]^d$, in particular the basis corresponding to the Karhunen-Loève expansion of the prior. The white noise model (2) is thus equivalent to observing

$$Y_k = \int_{[0,1]^d} \phi_k(x)dY_x = \langle \phi_k, f \rangle_{L^2} + \frac{1}{\sqrt{n}} \int_{[0,1]^d} \phi_k(x)dW_x =: \theta_k + \frac{w_k}{\sqrt{n}},$$

for $k = 1, 2, 3, \ldots$ and where $w_k \sim^{iid} N(0, \|\phi_k\|_{L^2}^2) = N(0, 1)$. The observations $(Y_k : k \geq 1)$ in the last equation and the original white noise model (2) are equivalent in the sense that each can be perfectly recovered from the other as we now explain. One can clearly obtain $(Y_k)$ as in the last display from the whole trajectory $(Y_x : x \in [0, 1]^d)$ by simply computing the integrals $(\int \phi_k(x) dY_x : k \geq 1)$. Conversely, suppose one observes $(Y_k : k \geq 1)$. For any $g \in L^2[0, 1]^d$, one can recover $Y_g$ in the second last display using the basis expansion $g = \sum_k \langle g, \phi_k \rangle_{L^2} \phi_k$ via

$$Y_g = \int g(x) dY_x = \int \sum_{k=1}^{\infty} \langle g, \phi_k \rangle_{L^2} \phi_k(x) dY_x = \sum_{k=1}^{\infty} \langle g, \phi_k \rangle_{L^2} Y_k.$$

Since this holds for arbitrary $g \in L^2[0, 1]^d$, one can reconstruct the process $(Y_g : g \in L^2[0, 1]^d)$ and thus the whole trajectory $(Y_x : x \in [0, 1]^d)$ as in (2). This shows these two representations are equivalent, and thus we may consider either as our 'data'.

Viewing the prior $\Pi$ through its series expansion (5), $\Pi$ can be viewed as a prior on the space of coefficients in the basis expansion of $\{\phi_k\}$ leading to the prior distribution $f = (\theta_k)_k \sim \otimes_{k=1}^{\infty} N(0, \lambda_k)$. Denoting by $P_f = P_f^n$ the distribution of the sequence representation (6), we have

$$P_f = \otimes_{k=1}^{\infty} N(\theta_k, 1/n).$$

Using Kakutani's product martingale theorem ([1], Theorem 2.7), the measures $(P_f : (\theta_k)_k \in \ell_2)$ are absolutely continuous with respect to $\otimes_{k=1}^{\infty} N(0, 1/n)$ with density

$$e^{\ell_n(f)} = \frac{dP_f}{dP_0} = \exp\left(\sqrt{n} \sum_{k=1}^{\infty} \theta_k Y_k - \frac{n}{2} \sum_{k=1}^{\infty} \theta_k^2\right) = \exp\left(\sqrt{n} \langle f, Y \rangle_{L^2} - \frac{n}{2} \|f\|_{L^2}^2\right),$$

so that $e^{\ell_n(f)}$ and $\ell_n(f)$ are the likelihood and log-likelihood of the model, respectively. Note that the likelihood is invariant to the choice of basis $\{\phi_k\}$, but the particular choice can (and will) provide a convenient representation.

Since the likelihood factorizes in terms of the coefficients $(\theta_k)_k$, a prior that makes the $(\theta_k)_k$ independent as in (5) will yield similar independence in the posterior. The posterior distribution is therefore conjugate and takes the form

$$f = \sum_{k=1}^{\infty} \theta_k \phi_k, \qquad \theta_k | Y_k \sim^{ind} N\left(\frac{n\lambda_k}{n\lambda_k + 1} Y_k, \frac{\lambda_k}{n\lambda_k + 1}\right),$$

where the exact form follows from standard one-dimensional conjugate computations for the normal likelihood $Y_k | \theta_k = N(\theta_k, 1/n)$ with normal prior $\theta_k \sim N(0, \lambda_k)$. This gives the form of the posterior (7) and its mean (8).