# OpenReview forum: "On the inability of Gaussian process regression to optimally learn compositional functions"
_NeurIPS.cc/2022/Conference — NeurIPS 2022 Accept_

### Official Review · Reviewer_SNzg · 2022-07-05

**Rating:** 7
**Confidence:** 3
**Soundness:** 4 excellent
**Presentation:** 4 excellent
**Contribution:** 3 good

**Summary:**

In the domain of frequentist guarantees of Bayesian nonparametric models, the paper exhibits a case where Gaussian processes learn slower than the minimax rate. This is of particular interest because for the same function class a recent paper showed that deep Gaussian do achieve the minimax rate. The sub-optimality of the GP is shown to be at least polynomial under very general assumptions. The function class in question is a type of generalized additive model. So together with the previous paper on deep GPs this is a formalization of the idea that deep methods do better when the "true" function is compositional in a certain particular sense. There is a relatively self contained exposition of the necessary background. The compositional function class considered does not easily fall into the domain of previous theoretical work lower bounding the rates of linear methods so providing a proof method is the main part of the technical novelty.

**Questions:**

I have no questions.

**Limitations:**

As far as I can see there is no potential negative societal impact of this work unless one was against science and mathematics in general. The scientific limitations are very clearly communicated and discussed leaving ample opportunities for follow up work.

**Strengths And Weaknesses:**

Clarity: I found the exposition of the results very clearly written considering that this is a notoriously challenging area of the literature. Work like this would often appear in a statistics journal and it felt like a particular effort had been made to make the work accessible to the NeurIPS theory community. Each non-trivial statement was carefully referenced. I hope to learn from such exposition in my own work. Make though mistake though, the work is still technically quite challenging for an average non-theoretical NeurIPS reader and the proof section is very heavy going I suspect for the big majority of potential readers. So be it.

Originality: There is a very clearly defined research question for the paper and this topic is to the best of my knowledge novel. Answering the question requires highly non trivial proof methods.

Quality: The expositional sections are of a high quality. I spent a significant amount of time on the paper and the proof method looks sensible to me but to carefully verify it would take more time than is available particularly given heavy reviewing loads this year. Given that similar work appears for instance in the Annals of Statistics where peer review takes significantly longer it is difficult to verify it to that level in this venue. It helps that there are two independent proof methods for Theorems 1 and 2 and that the result is not entirely unexpected.

Significance: The work is significant enough for publication in this venue in my opinion. At a broad level all work in this area is somewhat abstracted from practical reality. The work is in a sense "doubly" asymptotic since it requires both the reformulation in terms of a "statistically equivalent" SDE to be valid and then the number of data points to be sufficiently large on top of that. Of course chaining asymptotes is fine if this is your only goal but it does of course make it harder to understand the implications for numbers of data points and dimensionality encountered in practice. Similarly work in this area neglects computational constraints. After all both GPs and deep GP posteriors require approximation in practice. This is again fine as long as this is understood. More specifically to this paper the most general difference in rate (Theorems 1 and 2) is proven to be a polynomial and the level of difference is bounded as the dimension increases. It is tantalizing that the authors speculate that the more dramatic difference in rate proven under the special conditions of Theorem 3 may apply more generally. This obviously would increase the significance of the work but I appreciate how difficult such results are to obtain.

---

> ### Author Response · Authors · 2022-08-01
> **Reply to Reviewer SNzg**
>
> Thank you for the positive assessment of our work and kind words.

---

### Official Review · Reviewer_sPbe · 2022-07-08

**Rating:** 7
**Confidence:** 4
**Soundness:** 4 excellent
**Presentation:** 3 good
**Contribution:** 3 good

**Summary:**

This paper derives a novel lower bound for the GP posterior contraction rate when the true function has an additive structure. The result shows that for any GP in such scenario, the contraction rate is worse than the minimax estimation rate of deep GP. The authors further demonstrate that the contraction rate is even more suboptimal on a specific GP (e.g., Gaussian wavelet series). This is a purely theoretical work and there is no empirical study.

**Questions:**

- In line 113, when the minimax rate is introduced, I had to assume that this is the rate for deep GP prior based on the motivation of this work. Eq. (13) of [32] seems to suggest that the analysis is for general nonparametric regression with deep NN. Can you elaborate on the extension of this to GP with deep prior?
- Also [32] seems to suggest that the deep architecture has to satisfy certain conditions to achieve such rate. Can you provide a brief summarization of what those conditions entail, and whether they are realistic in practice? For example, in practice we also have to train the weights of the network, and I suppose a large architecture would affect the convergence rate.
- As a non-expert, I am quite curious if a contraction rate for deep GP has been lower bounded for general $f_0$ ? If so, how would the difference compare to the additive scenario?
- How do we arrive at the expectation in line 18 of the appendix?

**Limitations:**

The authors have not discussed negative societal impact of their work. However, since this is a purely theoretical work, I do not think there is any potential problem.

**Strengths And Weaknesses:**

This paper provides two original proving strategies to obtain the lower bounds of GP posterior and posterior mean contraction rates in the generalized additive function setting. The first strategy directly derives these lower bounds, whereas the second strategy reduces the regression to a one-sparse sequence model then lower bounds the minimax risk for its linear estimators. To the best of my ability, I have verified that both proofs are sound and original. Nonetheless, I acknowledge that I have taken some derivation steps for granted and my lingering questions are listed in the section below.

The paper is generally well presented and its high level ideas are quite easy to follow, although there are some small issues with the notations and technical clarity. For example:
- The symbol $L$ is overloaded to imply both Lipschitz constant and the $L^2$ space.
- In line 109, it seems to me that $f_i$ was overloaded to imply both the $i^{\text{th}}$ additive component of $f_0$ and later on $f_0(x_i)$ with $x_i$ being the $i^{\text{th}}$ observed input. I could be wrong, so I invite the authors to clarify this.
- The subscript $i$ in $x_i$ is also used to denote both the $i^{\text{th}}$ observed input and the $i^{\text{th}}$ dimension of an arbitrary input.

Overall, I believe the paper delivers a good theoretical contribution, and it is sufficient without empirical demonstration. However, it would be more compelling to see if the proposed theory aligns with practice, especially in settings that specifically construct the latent function $f_0$ to be of additive form, as assumed in this paper.

---

> ### Author Response · Authors · 2022-08-01
> **Reply to Reviewer sPbe**
>
> Thank you for the constructive suggestions and helpful comments. In reply to your comments:
>
> 1. The symbol $L$ is overloaded to imply both Lipschitz constant and the $L^2$ space.
>
> The Lipschitz constant has been changed to $\Lambda.$
>
> 2. In line 109, it seems to me that $f_i$  was overloaded to imply both the $i$-th additive component of $f_0$  and later on $f_0(x_i)$ with $x_i$ being the $i$-th observed input. I could be wrong, so I invite the authors to clarify this.
>
> Corrected, thanks.
>
> 3. The subscript $i$ in $x_i$ is also used to denote both the $i$-th observed input and the $i$-th dimension of an arbitrary input.
>
> The $x_i$ in the nonparametric regression has been renamed $X_i$ to clearly differentiate these cases.
>
> 4. In line 113, when the minimax rate is introduced, I had to assume that this is the rate for deep GP prior based on the motivation of this work. Eq. (13) of [32] seems to suggest that the analysis is for general nonparametric regression with deep NN. Can you elaborate on the extension of this to GP with deep prior?
>
> We now explicitly state and reference that this rate is attainable by suitably calibrated deep GPs.
>
> 5. Also [32] seems to suggest that the deep architecture has to satisfy certain conditions to achieve such rate. Can you provide a brief summarization of what those conditions entail, and whether they are realistic in practice? For example, in practice we also have to train the weights of the network, and I suppose a large architecture would affect the convergence rate.
>
> In [32] it is assumed that the depth of the network is of the order $O(\log n)$ with $n$ the sample size. It is known that for ReLU networks, this is necessary to get the optimal convergence rates for smooth regression functions. Moreover, [32] assumes a minimal width of the hidden layers and more importantly, the network sparsity ($=$ number of non-zero network parameters) needs to be of the order $n \times$minimax rate (up to $\log n$ factors).
>
> We prefer not to mention the details underlying the deep network architectures in [32], since we are focusing more on Bayesian approaches. We have instead added a sentence before Theorem 1 overviewing the hierarchical deep GP construction in [10], which attains the minimax rate. We note that the deep GP prior in [10] is selected for theoretical reasons and indeed may not reflect empirical practice.
>
> 6. As a non-expert, I am quite curious if a contraction rate for deep GP has been lower bounded for general $f_0$? If so, how would the difference compare to the additive scenario?
>
> No, we are not aware of any such results. Most proofs of lower bounds for GPs use specific properties of Gaussian measures (e.g. the form of the posterior mean) that do not extend straightforwardly (if at all) to deep GPs. Indeed, first upper bounds for contraction rates, which are generally much better understood, have only recently been obtained for deep GPs in [10].
>
> 7. How do we arrive at the expectation in line 18 of the appendix?
>
> In the line before, $w_k\sim^{iid} N(0,1)$ are the only random variables. We now use $E[w_k]=0$ and $E[w_k^2]=1$ together with the linearity of the expectation. This causes the terms $I$ and $III$ to disappear.

---

### Official Review · Reviewer_txpX · 2022-07-09

**Rating:** 7
**Confidence:** 5
**Soundness:** 3 good
**Presentation:** 3 good
**Contribution:** 3 good

**Summary:**

In this paper, the authors studied information-theoretic lower bounds of posterior contraction rate for Gaussian process regression with compositional assumptions. Specifically, if the true function is a generalized additive function, the posterior contraction rate of any zero-mean Gaussian process (irrespective of the choice of kernel) is strictly slower than the minimax rate. Even the posterior mean will be a suboptimal reconstruction and implies an uninformative uncertainty quantification. A sharper lower bound is also studied, showing that the performance under the Gaussian wavelet series priors suffers from the curse of dimensionality.

Overall, the paper is well written and easy to follow.

**Questions:**

The following questions may be considered by the authors.

1. How would the range ([0,1]) and the distribution of points x affect the theorem results? Since the points may not follow U [0,1] in real applications.

2. How would the feature extraction methods (neural networks, projections, etc.) affect the correctness of Theorem 1? As lots of works with high-dimensional points or multi-modal data will first project the data to a low-dimensional feature space with better expressiveness. If the dimension is lower than 3, according to Theorem 1, then the contraction rate can be accepted.




**Limitations:**

This paper concentrates on the zero-mean Gaussian process with sample data generating process and the underlying function with a special case of the compositional functions, it would be better to extend it to general mean function.

**Strengths And Weaknesses:**

Various proofs and frequentist assessments are provided to measure the speed of posterior contraction around the true regression function, which is a solid work that helps people to understand the theory underlying deep Gaussian process and the curse of dimensionality. For varied research backgrounds, it is recommended to add an extra paragraph to elaborate on the notations and mathematical operators used throughout the paper.

---

> ### Author Response · Authors · 2022-08-01
> **Reply to Reviewer txpX**
>
> Thank you for the constructive suggestions and helpful comments. In reply to your comments:
>
> 1. Add an extra paragraph to elaborate on the notations and mathematical operators used throughout the paper.
>
> We have added a notation paragraph in Section 2 (p2) explaining the main mathematical notations needed to understand the results of the paper, and provide more details when high-level mathematical objects are introduced. The proofs do require some additional notation, but we felt it best to define these locally since they are generally used in only a single place, and we wish to keep the main paper as unencumbered as possible.
>
> 2. How would the range ($[0,1]$) and the distribution of design points $x$ affect the theorem results?
>
> We believe that if the inputs are sufficiently evenly scattered on $[0,1]^d,$ the same results will hold. For instance, this should be true if the design point distribution $H$ (i.e $X_i\sim^{iid} H$) has a density that is bounded away from zero and infinity. We have added a paragraph in the Discussion (Section 3) on this, where we also mention the related problem of heteroscedastic (non-uniform) noise level. On the other hand, if the input distribution $H$ is for example discrete, we expect that the results will change (e.g. we may have to change the loss to $L^2(H)$ to reflect this - i.e. weight the loss by the probability distribution $H$). The domain range $[0,1]^d$ can be extended to any $[a,b]^d$ without changing the rates in the lower bounds (the constants will, however, depend on $a,b$).
>
> 3. How would the feature extraction methods (neural networks, projections, etc.) affect the correctness of Theorem 1? As lots of works with high-dimensional points or multi-modal data will first project the data to a low-dimensional feature space with better expressiveness. If the dimension is lower than 3, according to Theorem 1, then the contraction rate can be accepted.
>
> Excellent question! We believe that if one allows for general (arbitrarily non-linear) feature extraction methods, the optimal minimax rate should be attainable. The reason is that if one has a good (nonlinear) feature extraction scheme (e.g. the empirical basis coefficients with respect to a pre-selected basis) recovery of the full regression function $f$ can become rather straightforward (in the case of basis coefficients, one can just take the series estimator with respect to this basis).
>
> In particular, if one can learn the nonlinearity of the underlying function (in our case the compositional part), then the underlying function might be a `nice' function of the learned features. In this case, placing a GP as a function of the learned inputs no longer deals with a compositional function, and thus can potentially attain the optimal rate. If one constrains the feature extraction, e.g. to certain projections, similar results to ours may be true, but we are unsure. We have added a paragraph on this in the Discussion (Section 3).
>
> 4. This paper concentrates on the zero-mean Gaussian process with sample data generating process and the underlying function with a special case of the compositional functions, it would be better to extend it to general mean function.
>
> Since we are interested in lower bounds uniformly over a symmetric function class, centering the prior at a function $m\neq 0$ will intuitively be detrimental to recovering the function $-m$. In other words, the prior mass is now further away from the target function $-m.$ Thus within our framework, a non-zero prior cannot lead to faster uniform lower bounds.
>
> We agree that it would be interesting to work this out precisely, but it will make all our proofs significantly more technical and thus shift the focus away from the main proof ideas to more lengthy computations. For the sake of clarity, we  therefore prefer to stick to centered GPs.

---

### Meta-Review · Area_Chair_i4nP · 2022-08-25

**Recommendation:** Accept
**Confidence:** Certain

**Metareview:**

The reviewers unanimously agree that the theory here exhibiting a particular case where Gaussian process priors are inferior to deep Gaussian processes is interesting, and furthermore that the proof techniques themselves are novel. Indeed, reviewers had minimal or no substantial concerns about the paper, and most of the questions asked by reviewers txpX and sPbe read as simple follow up questions that the authors may choose to include discussion on.

**Award:**

No

---

### Decision · Program_Chairs · 2022-09-14

Accept